# An Integrated Analysis of Cashmere Fineness lncRNAs in Cashmere Goats

**DOI:** 10.3390/genes10040266

**Published:** 2019-04-02

**Authors:** Yuan Y. Zheng, Sheng D. Sheng, Tai Y. Hui, Chang Yue, Jia M. Sun, Dan Guo, Su L. Guo, Bo J. Li, Hui L. Xue, Ze Y. Wang, Wen L. Bai

**Affiliations:** 1College of Animal Science & Veterinary Medicine, Shenyang Agricultural University, Shenyang 110866, China; zhengyuanyuan07@163.com (Y.Y.Z.); ssda111@126.com (S.D.S.); huitaiyu0529@163.com (T.Y.H.); 15909826217@163.com (C.Y.); s541524030@163.com (J.M.S.); libojiang12@126.com (B.J.L.); huilingxue_2002@126.com (H.L.X.); 2Animal Genetic Breeding, Academy of Animal Husbandry Science of Liaoning Province, Liaoyang 130062, China; dan810308@163.com; 3Prosperous Community, Changshun Town, Huade 013350, China; 17824229545@163.com

**Keywords:** cashmere fineness, lncRNA, differently expressed genes, RNA-seq, lncRNA–targets

## Abstract

Animal growth and development are regulated by long non-coding RNAs (lncRNAs). However, the functions of lncRNAs in regulating cashmere fineness are poorly understood. To identify the key lncRNAs that are related to cashmere fineness in skin, we have collected skin samples of Liaoning cashmere goats (LCG) and Inner Mongolia cashmere goats (MCG) in the anagen phase, and have performed RNA sequencing (RNA-seq) approach on these samples. The high-throughput sequencing and bioinformatics analyses identified 437 novel lncRNAs, including 93 differentially expressed lncRNAs. We also identified 3084 differentially expressed messenger RNAs (mRNAs) out of 27,947 mRNAs. Gene ontology (GO) analyses of lncRNAs and target genes in cis show a predominant enrichment of targets that are related to intermediate filament and intermediate filament cytoskeleton. According to the Kyoto Encyclopedia of Genes and Genomes (KEGG) analysis, sphingolipid metabolism is a significant pathway for lncRNA targets. In addition, this is the first report to reveal the possible lncRNA–mRNA regulatory network for cashmere fineness in cashmere goats. We also found that lncRNA XLOC_008679 and its target gene, KRT35, may be related to cashmere fineness in the anagen phase. The characterization and expression analyses of lncRNAs will facilitate future studies on the potential value of fiber development in LCG.

## 1. Introduction

The cashmere goat is vital for the daily lives of some human societies, and it is famous for its excellent fiber production characteristics. However, cashmere is getting coarser every year. Researchers have sought ways of improving the quantity and quality of cashmere, and therefore, the fineness and quality of cashmere have attracted much attention [1,2]. The Liaoning cashmere goat (LCG) and Inner Mongolia cashmere goat (MCG) are excellent domestic breeds with large cashmere yields, and each of these breeds has its own advantages. MCG is known as a “soft gold” breed, with the production of soft and fine cashmere fiber, while LCG has higher cashmere production and coarse fiber diameter. At present, research on changing the cashmere fineness of LCG is a main priority.

Long non-coding RNA (LncRNA) transcripts are longer than 200 bp, and in recent years they have been proven to play a major role in many biological processes, such as cell growth and proliferation [3], developmental and pathological processes [4], the regulation of intracellular processes [5], and in transcriptional regulation [6,7,8]. Interactions with messenger RNAs (mRNAs) or microRNAs (miRNAs) impact on the functions of lncRNAs [9]. LncRNAs that play a role in the hair growth of cashmere goats have been identified [10]. A total of 1108 lncRNAs were identified that may affect hair follicle cycling in cashmere goats, some of which were associated with hair follicle development by adjacent or co-expression [11]. RNA sequencing (RNA-seq) indicates that in hair follicles, 91 lincRNAs were significantly expressed; moreover, in goat skin, 492 lincRNAs and 2350 protein-coding genes were differentially expressed [12]. Research demonstrated that there were 7000 transcripts that are differentially expressed in cashmere skin during the anagen and telogen phases [13]. Overexpression of lncRNA-000133 resulted in a significant increase in the relative expression of ET-1, SCF, ALP, and LEF1 in dermal papillary cells [14]. Therefore, we speculated that the lncRNAs have a critical role in the fineness of cashmere goat hair.

In this work, we investigated the regulation mechanism of cashmere fineness, for an in-depth understanding of the regulatory networks of related genes. We aim to explore the functions of lncRNAs and target genes in LCG. We applied RNA-seq to examine differentially expressed (DE) mRNAs and lncRNAs in LCG and MCG skin, and the key candidate genes and lncRNAs related to cashmere fineness were found by gene ontology (GO) and Kyoto Encyclopedia of Genes and Genomes (KEGG) enrichment analyses. The concealed mechanisms in molecular regulation underlying cashmere fiber growth have not been absolutely determined [15]. This study provides a useful reference for further understanding of the relationship between lncRNAs and cashmere fineness, and it will contribute to knowledge on the development of cashmere traits.

## 2. Materials and Methods

### 2.1. Sample Preparation and Ethics Statement

We collected scapular skin samples from three LCGs and three MCGs. The sample size was 2 cm^2^ of female adult within the growth period, which had the same feed conditions. We used procaine for local anesthesia to reduce animal pain. Skin tissues were frozen in liquid nitrogen and stored at −80 °C until RNA was isolated. In addition, the three female adult LCGs with coarse-type skin in anagen stage and three female adult LCGs with fine-type skins in anagen stage were collected for qRT-PCR. All goat experimental procedures used in this study were approved and conducted according to the guidelines by the Laboratory Animal Management Committee of Shenyang Agricultural University.

### 2.2. RNA Extraction, Library Preparation, Sequencing, and Quality Analysis

Total RNA was extracted from cashmere goat skin tissues by using a Takara kit (Dalian, China) according to the manufacturer’s instructions. The contamination and degradation of RNA were detected by 1% agarose gels. The total RNA was stored at −80 °C for future use. The total RNA amount of each sample was equal to 3 μg RNA as the input material for RNA library preparation. Firstly, ribosomal RNA was removed using the Epicentre Ribo-zero rRNA Removal Kit (Epicentre, Madison, WI, USA), and the rRNA was cleaned up by ethanol precipitation. Subsequently, in total, two libraries were generated from the rRNA-depleted RNA using the NEBNext Ultra Directional RNA Library Prep Kit for Illumina (NEB, Ipswich, MA, USA) following the manufacturer’s recommendations. Two library preparations were sequenced on an Illumina Hiseq 2000 platform. The 150 bp pair-end reads were generated. FastQC was used to evaluate the quality of the raw read data [16]. Reads exceeding 10% unknown bases, reads containing adapters, and reads with low-quality bases were removed by FastQC. Simultaneously, a check on clean reads (Q20, Q30, and GC content) was performed. All later analysis was based on the high-quality data.

### 2.3. Identification of lncRNAs

LncRNAs were identified according to the following workflow. Firstly, we used Tophat2 [17] to map the reads to the reference genome [18]. The mapped reads of each sample were assembled by Cufflinks [19]. Then, considering the three basic conditions (length ≥ 200 nt, exon count ≥ 2, and read coverage ≥ 3) for transcripts, the unqualified transcripts were discarded. The class-code information originating from Cuffcompare was used to screen the candidate transcripts. Finally, transcripts that passed the protein-coding-score test were retained using CNCI [20], CPC [21], PFAM [22], and phyloCSF [23]. CNCI software was used with the default parameters. CPC was mainly used to detect the range and quality of the open reading frame (ORF) in transcripts, to discover sequences in known protein databases, and to identify coding and non-coding transcripts. For PFAM, each transcript was translated into all three possible frameworks, and the Pfam Scan was used to identify the presence of any known protein family domains that were recorded in the Pfam database (http://pfam.xfam.org/). From the overlapping results of the four software programs, those transcripts that were candidate lncRNAs with no coding potential were noted for further analysis.

### 2.4. Expression Profiling and Analysis of Differential Expression

Cufflinks was used to calculate the fragments per kb per million reads (FPKM) for both lncRNAs and coding genes in each sample. Differentially expressed mRNAs and lncRNAs between the two libraries were identified by edgeR with raw read counts [24]. We used q-value ≤ 0.05 and |log_2_FoldChange| ≥ 1 as thresholds to evaluate the statistical significances of differences in mRNAs and lncRNA expression.

### 2.5. LncRNA Target Gene Prediction and Enrichment Analysis

In order to explore the functions of lncRNAs, we predicted the target genes of lncRNAs in cis. The coding genes of the sequences that were, respectively, 10 kb and 100 kb upstream and downstream of the lncRNAs, were retrieved. All differentially expressed mRNAs and target genes of lncRNAs were mapped to GO terms from the Gene Ontology Consortium, and the number of genes for each GO term was calculated according to the GOseq R package. GO terms with corrected *p*-values < 0.05 were considered to be significantly enriched by DE genes. KEGG was used to enrich and identify the significant metabolic pathways or signal transduction pathways (http://www.genome.jp/kegg/).

### 2.6. Validation of lncRNA Expression by Quantitative Real-Time PCR

In order to find the difference of expression in coarse-type and fine-type LCG, lncRNAs were randomly selected to confirm the expression levels, and differences in the expression of lncRNAs in skin were detected by qRT-PCR. Primers for quantitative PCR were designed with Primer 5.0. Each type of cashmere goat breed included at least three samples, and all reactions were carried out three times for each sample. The GAPDH gene was used to standardize the expression level of quantitative real-time PCR [14], and to calculate the relative expression using the 2^−ΔΔCt^ method [25].

### 2.7. Construction of the lncRNAs–Targets Network

We constructed a lncRNAs–targets network combining 93 instances of differential expression of lncRNAs with their corresponding target genes, and nine validated lncRNAs and their correlated mRNAs. Networks of lncRNAs and target genes were created by Cytoscape 3.5.1 software [26], and the target genes of lncRNAs were predicted in cis.

## 3. Results

### 3.1. Identification of mRNA and lncRNAs in LCG and MCG

A total of 136,441,946 and 154,106,016 raw reads were generated from RNA-seq data obtained for skin samples of the LCG and MCG species, respectively (Table 1). Moreover, 128,989,956 (LCG) and 146,038,848 (MCG) clean reads were obtained for further analysis after filtering out of poly-N > 10%, adapter-polluted reads, and low-quality reads. The mapping ratio of clean reads in LCG and MCG were 86.25% and 84.37%, respectively. The total mapped reads were compared to each chromosome on the genome of goat; there was an obvious difference in the density distributions of reads on chromosomes between LCG and MCG. Finally, 437 assumed non-coding transcripts were retained by using CNCI, CPC, PFAM, and phyloCSF software (Figure 1), including 306 lincRNAs, 101 intronic lncRNAs, and 30 anti-sense lncRNAs. In addition, 27,947 mRNAs were identified.

### 3.2. Characteristics of mRNAs and lncRNAs

In our study, we found that the characteristics of lncRNAs were similar to previous studies [27]. We analyzed the expression level between lncRNAs and protein-coding genes, and the results showed that the expression levels of lncRNAs were lower than protein-coding genes (Figure 2A), and that protein-coding genes were more conservative than lncRNAs, resulting in a cumulative distribution curve of mRNA and lncRNA conservation scores (Figure 2B). The sequence results of lncRNAs showed a certain degree of site conservation among the species, with regard to the mRNAs. In our data, most of lncRNAs tended to be shorter in length, and they contained less exons than mRNAs (Figure 2C,D), and the lengths of ORFs in the lncRNAs were shorter than those of the mRNAs (Figure 2E).

### 3.3. Differential Expression Analysis between lncRNAs and mRNAs

We found 93 lncRNAs and 3084 mRNAs differentially expressed between LCG and MCG skin tissues (q-value ≤ 0.05 and |log_2_FoldChange| ≥ 1). Among the total of 3177 differentially expressed lncRNAs and mRNAs, 2119 were up-regulated and 1058 were down-regulated (Figure 3). To further explore the potential functions of lncRNAs, we performed a clustered heat map on the differentially expressed genes. Our results included 83 lncRNAs that corresponded to 104 mRNAs within a range of 10 K, as well as 303 lncRNAs that represented 1033 mRNAs within a range of 100 K. The basic principle of the cis function prediction was that the functions of lncRNAs were related to the adjacent protein coding genes. The top 10 differentially expressed lncRNAs and their potential target genes in LCG and MCG are shown in Table 2. Ten down-regulated genes and eight up-regulated genes are shown in Table 3, which may be involved in the physiological characteristics of hair growth. We performed an analysis on the target genes of lncRNAs and differentially expressed mRNAs, a total of 149 common genes were obtained, including KRT35, KRT32, and NFKBIA. KRT and KRTAP were related to the hair shaft, and they are important components of cashmere fiber.

### 3.4. GO and KEGG Functional Enrichment Analyses

In order to confirm the functions of DE lncRNAs, we performed GO term and KEGG pathway enrichment analyses, based on their regulated target mRNAs. These genes were enriched for 990 GO terms, of which 563, 137, and 290 were enriched in the term biological process, cellular component, and molecular function, respectively. LncRNA target genes enrichment to differential expression GO terms is shown in Table 4. The top 10 significantly enriched terms were as follows (Figure 4): Intermediate filament, intermediate filament cytoskeleton, viral capsid, sequence-specific DNA binding, virion part, drug transmembrane transport, drug transmembrane transporter activity, epidermis development, skin development, and virion. DE mRNAs were significantly enriched for 174 terms of GO analysis, including 65 terms for molecular function, 33 terms for cellular component, and 76 terms for biological process. Interestingly, lncRNA target genes and predicted mRNAs had overlapping results in the GO analysis, such as intermediate filament and intermediate filament cytoskeleton.

A total of 16 pathways were significantly enriched by using the Kyoto Encyclopedia of Genes and Genomes enrichment analysis (*p* < 0.05) (Figure 4B), and the top 10 pathways were related to α-linolenic acid metabolism, linoleic acid metabolism, taste transduction, fat digestion and absorption, ether lipid metabolism, systemic lupus erythematosus, alcoholism, glycerophospholipid metabolism, arachidonic acid metabolism, and histidine metabolism. Twenty-four pathways were significantly enriched, based on DE mRNAs. These results suggested that lncRNAs may act on their target genes to regulate the onset of the fineness of cashmere.

### 3.5. Validation of lncRNAs by qRT-PCR

To verify the reliability of the RNA-sequencing results, nine differentially expressed lncRNAs between coarse and fine LCGs were randomly selected to validate their expression levels, by using qRT-PCR (Figure 5). The results of the qRT-PCR displayed an almost consistent degree of high expression of lncRNAs for coarse-type LCG; these lncRNAs may regulate cashmere thickening and they may have positive correlations with cashmere fineness. This provides the basis for future functional verifications of these lncRNAs.

### 3.6. Networks of lncRNAs–mRNAs

LncRNAs can regulate mRNAs by targeting their functions; therefore, we constructed a co-expression network based on DE lncRNAs and their target genes (Figure 6A). Figure 6B is a network of validated lncRNAs and their correlated mRNAs. Hair follicle fibers are mainly composed of keratin and keratin-associated proteins, which determine the quality of the fibers, and they are related to villi growth. Based on the analysis of the lncRNAs for follicle formation and their surrounding genes, we found that lncRNAs regulate the transcription of adjacent genes, and the corresponding target genes are decreased in expression.

## 4. Discussion

At present, the fineness of cashmere is becoming thicker year by year, and good quality cashmere products are obviously becoming rarer. Reducing the thickness of cashmere is an important issue. RNA-seq allows large-scale data production via high throughput sequencing, which has helped us to detect transcripts with low abundances, identify novel transcript units, and investigate their differential expression between diverse samples [28,29]. Moreover, RNA-seq data can be used for feasible large-scale expression studies as well as for discovering a class of biologically significant RNA transcripts, and characterization of lncRNAs has been promoted by RNA-seq. However, little is known about lncRNAs for the fineness of cashmere goat skin.

In this study, we identified the differential expression of 93 lncRNAs and 3084 mRNAs. Consistent with similar studies on different organisms, the characteristics of lncRNAs are more conservative, as they have shorter sequence lengths, fewer exon numbers, and lower expression levels compared to those of mRNAs [30,31]. The nine lncRNAs that were randomly selected to be used for validating the RNA-seq results in different types of LCG, allowed us to come to the following findings. XLOC_010430, which was the most differentially expressed sequence between LCG and MCG skins, as indicated by RNA-seq analysis, had a high level of expression in coarse-type LCG. LncRNA XLOC_008679 may play an important role in cashmere fineness in the anagen phase, the high expression of XLOC_008679 in fine-type LCG was verified by qRT-PCR, and the differentially expressed gene KRT35 was predicted as the target gene of XLOC_008679. KRT35 is the basic building block of human hair [32]. Studies have shown that KRT and KRTAP are major structural proteins of the hair fiber and sheath, and their contents are important for fleece quality and cashmere goat hair morphology [33,34]. Through cDNA sequences, the transcripts of KRT40, KRT82, and KRT84 were identified as being only expressed in the fiber cuticle; KRT32, KRT35, and KRT85 were present in the cuticle and fiber cortex [35].

The roles of some lncRNAs in cashmere growth have been reported in previous studies; from the increasing volume of data, lncRNAs are involved in the periodic growth of velvet [36]. For instance, hair follicle morphogenesis is probably regulated by the lncRNAs TCONS_00255106 and TCONS_00206163 [37]. Compared with the telogen and catagen phases, the relative expression of lncRNA-H19 is significantly higher during secondary hair follicle anagen, which suggests that lncRNA-H19 transcription may play a role in the formation of cashmere goat hair [38]. Under certain conditions, lncRNAs may be involved in regulating cashmere goat hair follicle circadian rhythms, and lncRNAs that are mediated by melatonin may have major control over the pluripotency of stem cells [37]. A prior study has reported that lncRNAs LOC102190274, LOC108635596, LOC108635657, LOC108636746, LOC108635658, LOC102188339, LOC108635659, and LOC108635656 are potential regulators of cashmere growth [39]. Meanwhile, there are a few lncRNAs that are expressed specifically in the single developmental stage of cashmere cycling in the telogen phase, including Lnc_00092, Lnc_000183, Lnc_000406, and Lnc_000559, while Lnc_000173 shows an indication that these lncRNAs could regulate cashmere cycling by their spatio-temporal expression during the anagen phase [11].

Increasing numbers of studies have shown that non-coding RNAs play an important role in regulating development [40]; however, how lncRNAs regulate cashmere fineness remains unknown. Here, we predicted the potential functions of the identified lncRNAs by using cis methods; a total of 1037 target genes were detected for all the identified lncRNAs. Target genes KRT32, KRT35, KRT36, and KRT38 of XLOC_008679 and target genes PSMA6, NFKBIA, and KIAA0391 of XLOC_011060 were considered as key candidate genes for cashmere fineness [35,41]. Differentially expressed lncRNAs might be considered as potential candidate genes for further study on the molecular mechanisms of hair follicle morphogenesis; the interactions of lncRNA XLOC2437 and COL6A6 can regulate and reduce the deposition of collagen VI α6 chain in sheep skin by positive feedback, thus inhibiting skin fibrosis and promoting the formation and deposition of the placenta [36]. The lncRNAs LNC_000972, LNC_000503, and LNC_000881 may regulate hair follicle cycling because of their proximities to the genes for WNT3A, HOXC13, and MSX2, respectively [11].

There are 149 genes that overlap between the target genes and mRNAs in our study, and they are mainly from the keratin family and keratin-associated proteins. Previous studies have shown that KRT and KAP are main structural proteins of hair fibers and sheaths, and that their contents are also important for fleece quality [42]. The keratin family consists of cytoskeletal proteins with hierarchical structures, which can be arranged in filaments and function as dimer subunits [43]. Keratins have plentiful intermediate filaments, which can be found in skin, hair, and nails [44,45]. They can regulate mechanical stresses across the cell in skin, which can have an influence on the skin’s elastic properties [46,47]. Thus, we can make an assumption that keratin-associated proteins are responsible for the formation of the hair shaft and the alternation of hair structure and diameter [35]. From this point, keratin-associated protein genes may have a more important function for the structure of cashmere fibers. Keratin proteins and keratin-associated proteins are strongly associated with hair growth: KRT38, KRT4, KRTAP15–1, KRTAP13.1, and KRTAP3–1 were involved in the construction of hair, and they were more highly expressed in the anagen phase than in the catagen stage [39]. It has been identified that KRTAP11-1, TCHH, CALML5, FABP4, and FABP5 in Han sheep have quite different expression effects between wool and cashmere. KRT6A, KRTAP11-1, KRT38, KRT23, LAMP2, DSC2, DSG3, LOC102185652, LOC102176161, and LOC102175613 show differences in effects between guard hair and cashmere in cashmere goats. Simultaneously, the expression of KRTAP11-1 in fine fiber-type cashmere samples is higher than in wool samples [48]. Through RNA-FISH, CSDC2 expression was shown to be high within keratinocytes of the epidermis and fibroblasts of the dermis, and weak in the hair shaft [49]. Among the different Dsg isoforms, DSG4 was the only isoform that is highly expressed in the hair cortex [50]. Other data demonstrate that in the presence of Smad4, BMP-signaling participates in the transcriptional regulation of DSG4, thus, Smad4 loss-associated DSG4 depletion contributes, at least in part, to hair follicle degeneration in Smad4-deficient skin [51]. HOXC8 could be involved in the control of hair fiber growth and development [52]. HGF has been proven to be an effective regulator of hair growth, and it may contribute to development and growth cycles in hair follicles [53]. For the differential impact of MSP on hair follicles, an investigation to check whether it is involved in the modulation of hair growth was performed [54]. BMP4, as a candidate molecule identified in the telogen phase, is one of the growth regulators that are expressed in the dermal papilla [2,55]. Target genes of lncRNAs, such as SOX9, GATA3, BMP2, BMP4, and HOXC13, may be involved in the hair follicle cycle [11,39,43], as could be KRT35, KRT84, and DSG4, which are genes that show differential expression in this area [38]. TCHH, EDAR, GATA3, PRSS53, WNT10A, and OFCC1 have been shown to be related to human hair shape [56,57,58,59,60]. GWAs of the cashmere fiber trait revealed that AKT1, ALX4, NT-3 growth factor receptor, and HK1 were related to cashmere fineness in hair follicles, and that POLD2, PSMA2, RYR3, VPS39 (TLP), and STARD9 were related to cashmere fineness in skin [61], which were also found in our data. Lnc-AHNAK2-7 may be related AKT1 in the MAPK signaling pathway [62].

In our study, GO enrichment analysis of target genes showed clustering mainly within three terms (*q*-value < 0.05): Intermediate filament, intermediate filament cytoskeleton, and viral capsid. All of the target genes of XLOC_008679 were linked to intermediate filament and intermediate filament cytoskeleton GO terms, which have been proven to be related to the growth of cashmere. Differentially expressed mRNAs were also enriched in intermediate filament and intermediate filament cytoskeleton terms. Other studies have shown that DE genes and targets are enriched in the intermediate filament [39]. Thus, intermediate filaments may play an important role in cashmere fineness. Systemic lupus erythematosus, and the overlap pathway between lncRNA target genes and differentially expressed mRNAs, may also be involved in cashmere fineness. Research on the molecular mechanisms that control hair follicle cycling, especially the relevant signaling pathways, has advanced over the past 10 years [63,64]. The PPAR pathway may play a role during the start of secondary hair follicle development [36]; target genes SORBS1 and UBC were enriched to this pathway in our data. Wnt signaling is needed for the establishment of hair follicles, and it is up-regulated at telogen and facilitated into antigens, which mainly results in the activation of bulge stem cells for the promotion of warding hair formation [65]. We found differentially expressed genes WNT11, BAMBI, WIF1, PPARD, and MMP7 were also enriched to the Wnt signaling pathway. Target gene NFKBIA of XLOC_011060 and XLOC_011319 was enriched to the NF-kappa B signaling pathway, which, it was reported, may regulate hair follicle induction. The EDA/EDAR/NF-kB pathway is important for the development of hair follicles and epidermal appendages [66,67,68,69]. The lncRNA MTC promotes the regulation of hair follicle development and cashmere growth by activating NF-κB signaling [41]. We think that NF-kappa B signaling pathway may be related to cashmere fineness.

## 5. Conclusions

Target genes of differentially expressed lncRNAs and mRNAs were enriched in intermediate filament and intermediate filament cytoskeleton GO terms. This suggests that they may be key terms for cashmere fineness. We then verified nine differentially expressed lncRNAs in LCG coarse-type and fine-type goats by qRT-PCR analysis. We found that lncRNAXLOC_008679 and its target gene, KRT35, may be related to cashmere fineness in LCG. Our results clearly demonstrate that lncRNAs may be essential for regulating cashmere fineness in LCG.

## Figures and Tables

**Figure 1 genes-10-00266-f001:**
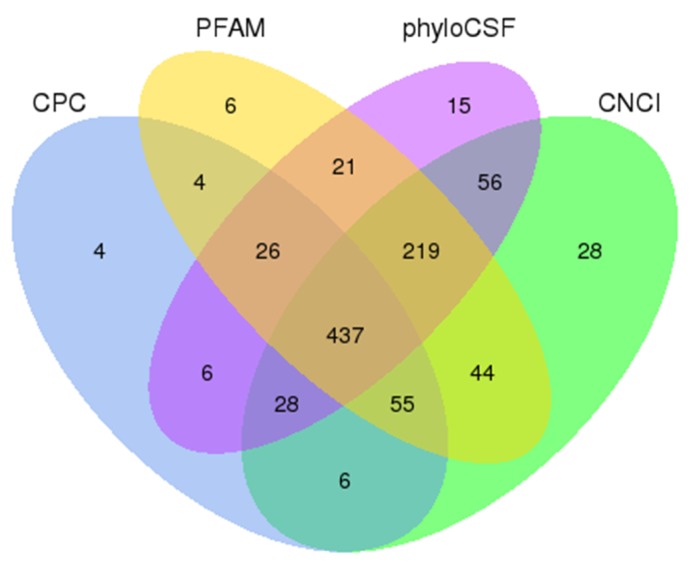
Identification of long non-coding RNAs (lncRNAs). The overlapping of noncoding transcripts shared by four software programs. The sum of the numbers in each large circle represents the total number of noncoding transcripts that are detected by the software.

**Figure 2 genes-10-00266-f002:**
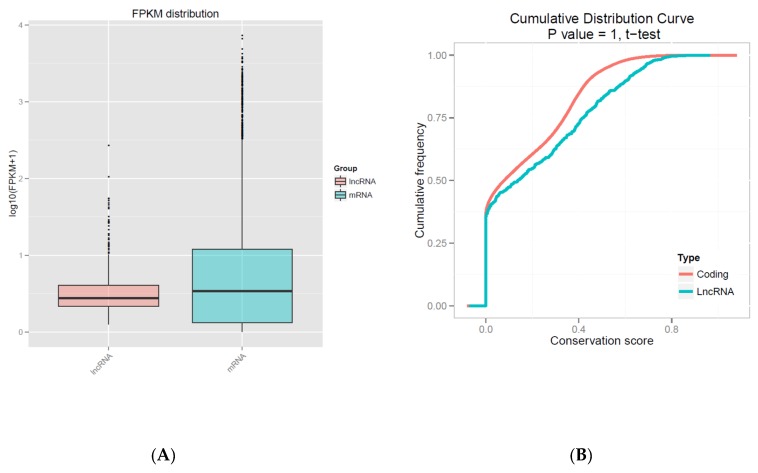
Characteristics of lncRNAs. (**A**) Expression-level analysis for messenger RNAs (mRNAs) and lncRNAs. (**B**) Conservative analysis of sequences in mRNAs and lncRNAs. (**C**) Length distribution of mRNAs (red) and lncRNAs (blue), unit of the length is bp. (**D**) Exon number distribution for mRNAs and lncRNAs. (**E**) Open reading frame (ORF) length distribution for mRNAs and lncRNAs.

**Figure 3 genes-10-00266-f003:**
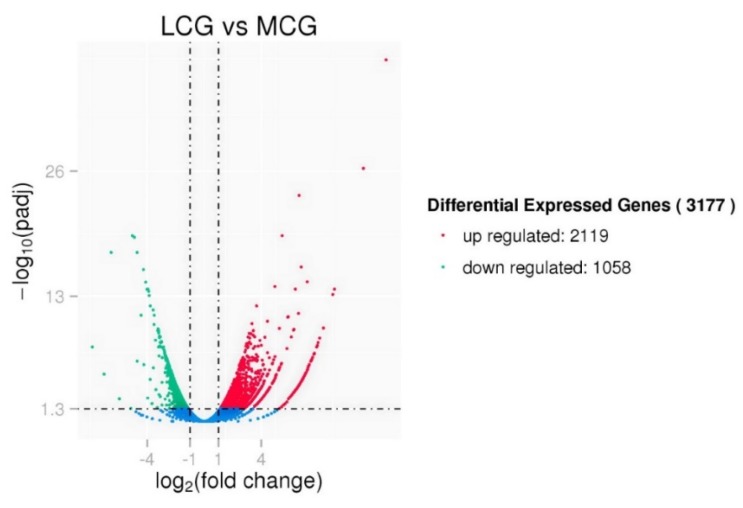
Volcano plot of differentially expressed genes in Liaoning cashmere goats (LCG) and Mongolia cashmere goats (MCG).

**Figure 4 genes-10-00266-f004:**
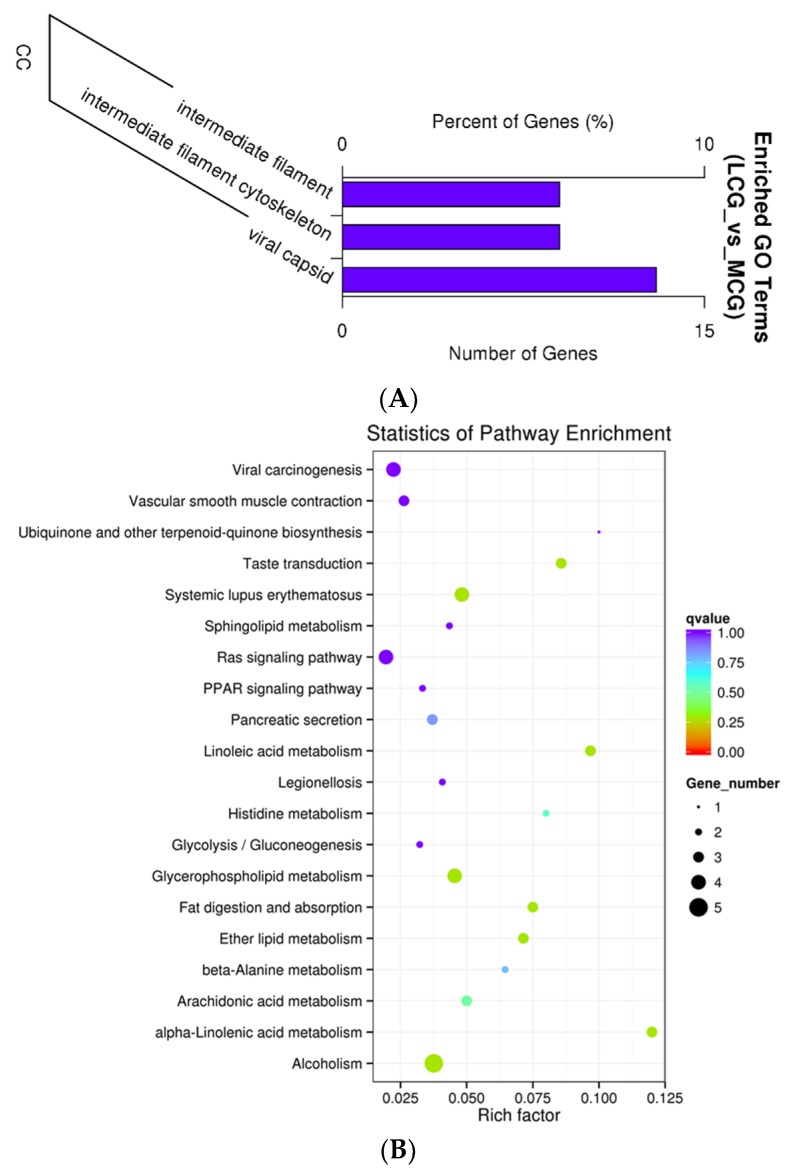
Target genes enrichment analysis. (**A**) GO enrichment analysis for target gene functions of predicted lncRNAs (CC: Cellular component). (**B**) Kyoto Encyclopedia of Genes and Genomes (KEGG) annotation for the target gene functions of predicted lncRNAs. The size of the dot indicates the number of target genes in the pathway, and the color of the dot corresponds to different q-value ranges.

**Figure 5 genes-10-00266-f005:**
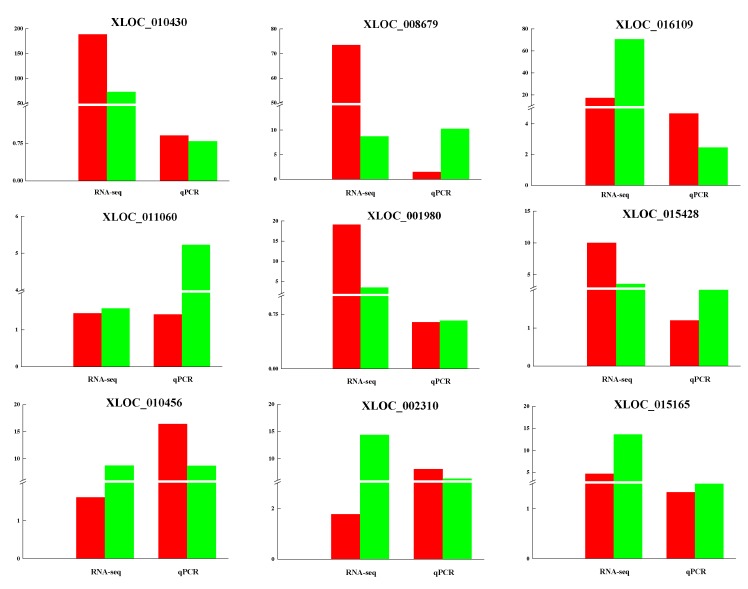
qRT-PCR validation of lncRNAs. RNA sequencing (RNA-seq) was used in verifying the expression of LCG and MCG, red: LCG, green: MCG. qPCR validation compared coarse-type and fine-type LCG, red: Coarse type, green: Fine type.

**Figure 6 genes-10-00266-f006:**
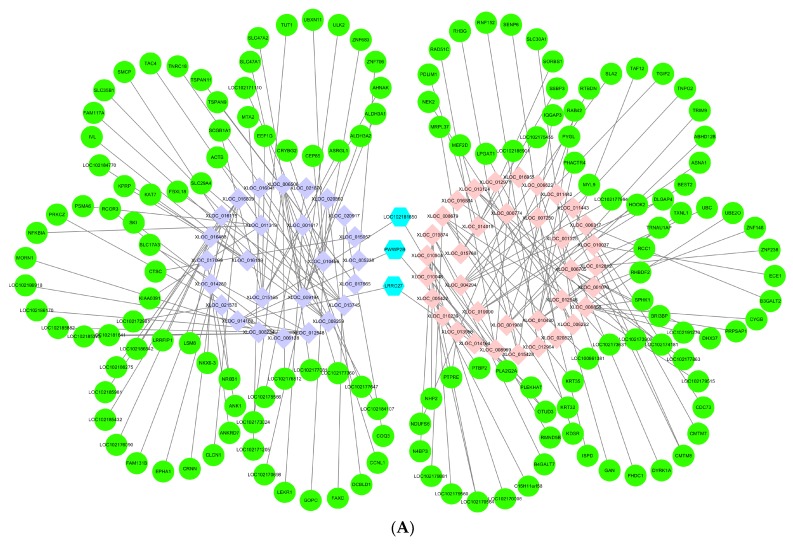
The network of differentially expressed lncRNAs. (**A**) Network of differentially expressed lncRNAs and target genes. Light pink: Up-regulation gene, lilac colour: Down-regulation gene, green: Target gene, light blue: Co-up-regulated and down-regulated target gene. (**B**) Network of qRT-PCR verification for lncRNAs and target genes. Light pink: Up-regulation gene, lilac colour: Down-regulation gene, green: Target gene.

**Table 1 genes-10-00266-t001:** Summary of reads mapping to the goat reference genome.1

Samples	LCG	MCG
Raw reads	136,441,946	154,106,016
Clean reads	128,989,956	146,038,848
Mapped reads	111,249,567	123,211,813
Mapping rate	86.25%	84.37%
Uniquely mapped reads	107,287,774	117,506,303
Unique mapping rate	83.18%	80.46%

**Table 2 genes-10-00266-t002:** Differentially expressed lncRNAs with target genes in LCG and MCG.

Accession No.	FPKM_LCG	FPKM_MCG	*p*-Value	Target Genes
*XLOC_010430*	188.553	73.0454	0.001316309	*TRNAU1AP; RCC1; TAF12; RAB42; PHACTR4*
*XLOC_001980*	19.1252	3.50528	4.68 × 10^−8^	*PYGL; ABHD12B; TRIM9*
*XLOC_016109*	17.2188	70.6548	2.98 × 10^−6^	*LOC102184770; KPRP; SMCP; IVL;*
*XLOC_008679*	84.16022883	10.1007685	1.92 × 10^−11^	*KRT32; KRT35; KRT36; KRT38;* *LOC102179515; LOC102179881* *;* *LOC100861381*
*XLOC_008969*	49.1703	2.20119	1.22 × 10^−14^	*LOC102186901; RAD51C*
*XLOC_016115*	6.23184	55.5671	1.81 × 10^−11^	*LOC102186542; CRNN; LOC102176090*
*XLOC_020822*	12.6	4.6006	0.004947	*SUMO1*
*XLOC_015428*	10.0065	3.5299	0.00064	*IQGAP3; RHBG; MEF2D*
*XLOC_015165*	4.64825	13.617	0.000989863	*SCGB1A1; TUT1; ASRGL1; AHNAK;* *MTA2; EEF1G*
*XLOC_011319*	1.15849	2.77598	0.005641	*PSMA6; NFKBIA; KIAA0391*

**Table 3 genes-10-00266-t003:** Differentially expressed genes in LCG and MCG.

Gene Name	FPKM_LCG	FPKM_MCG	log_2_FoldChange	*p*-Value
*LOC102183488*	12,694.8	1922.63	−2.62355	1.90 × 10^−9^
*LOC102177231*	7911.92	1767.08	−2.14703	5.15 × 10^−7^
*TCHH*	6322.89	1638.67	−1.93589	5.11 × 10^−6^
*LOC102185436*	4516.72	767.495	−2.54179	4.75 × 10^−9^
*LOC100861381*	4044.28	858.502	−2.22033	2.25 × 10^−7^
*KRTAP11-1*	3930.26	697.121	−2.47797	1.06 × 10^−8^
*KRTAP3-1*	3190.44	431.082	−2.86825	7.23 × 10^−11^
*LOC100861181*	3186.97	162.853	−4.27184	8.52 × 10^−20^
*LOC102184223*	2111.34	361.071	−2.45445	1.40 × 10^−8^
*KRT35*	994.396	168.398	−2.546514249	4.58 × 10^−^^9^
*HOXC13*	135.246	26.9767	−2.310349019	9.09 × 10^−^^8^
*KRT82*	450.878	68.8984	−2.270889688	1.30 × 10^−^^7^
*LOC100861183*	1269.99	80.5535	−3.957981688	1.59 × 10^−^^17^
*TCHHL1*	196.326	45.1137	−2.10699655	8.45 × 10^−7^
*DSG4*	252.01	59.1967	−2.075452034	1.18 × 10^−6^
*BMP2*	17.1057	4.94124	−1.775970627	4.34 × 10^−5^
*LOC102188576*	58.099	732.179	3.672081	1.12 × 10^−15^
*ACMSD*	21.1133	213.701	3.353863	1.07 × 10^−13^
*CCDC80*	34.9971	307.292	3.148606	1.61 × 10^−12^
*RCN1*	28.2099	236.021	3.079448	4.55 × 10^−12^
*COL3A1*	802.38	4,412.04	2.473234	1.11 × 10^−8^
*LOC102189356*	53.903	231.887	2.120764	7.69 × 10^−7^
*LOC102187909*	193.598	829.073	2.112533	7.61 × 10^−7^
*COL1A2*	834.656	2821.44	1.771336234	2.74 × 10^−5^

**Table 4 genes-10-00266-t004:** Gene ontology (GO) term of cashmere fineness.

GO Term	lncRNA	Up-Target Gene	Down-Target Gene
intermediate filament	*XLOC_008679*		*LOC102179515; KRT32; KRT36; KRT35; LOC102176457; LOC102176457; KRT38; LOC100861381; LOC102179881*
intermediate filament cytoskeleton	*XLOC_008679*		*LOC102179515; KRT32; KRT36; KRT35; LOC102176457; LOC102176457; KRT38; LOC100861381; LOC102179881*
viral capsid	*XLOC_008679*		*KRT35; KRT36; LOC100861381; LOC102179515; KRT32; KRT38*
*XLOC_019874*		*RMND5B*
*XLOC_020860*		*GOPC*
*XLOC_010504*	*LOC102170006; PLA2G2A*	
*XLOC_001617*	*LEKR1*	
*XLOC_015428*		*IQGAP3*
*XLOC_006506*	*LOC102177360*

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
