# Peer review of "An Integrated Analysis of Cashmere Fineness lncRNAs in Cashmere Goats"

_genes, 2019, doi:10.3390/genes10040266_

Round 1
Reviewer 1 Report
The work has used the RNA-seq approach to investigate issues related to long non-coding RNAs (lncRNAs) and mRNA expressions in two species of goats. The authors found relevant information about key lcnRNA and target genes and the findings are relevant to a better comprehension of differentially expressed genes involved with the quality of the cashmere. The paper is well written, though some suggestions were proposed to improve it even more. However, the methodology description needs to be improved as well as the end of the discussion. Comments and suggestions are in the attached manuscript file.

Author Response
1. delete RNA
I have deleted RNA.
2. include "approach" after (RNA-seq)
I have added “approch” after (RNA-seq).
3. replace "elucidate" with "investigated"
I have replaced it.
4. replace "will provide" with "provides"
I have replaced it.
5. It is not clear if the samples were obtained from six different individuals for each line/species, or tree, or even a different number. Please, clarify the number of individuals sampled for each LCG and MCG goat, including the N for both coarse and fine lines
E.g. We collected scapular skin samples from N LCG (N coarse and N fine) and N MCG cashmere goats. The sample size was 2 cm2 of female adults within the growth period, which had same feed conditions.
I have changed it to that “We collected scapular skin samples from three LCGs and three MCGs. The sample size was 2 cm2 of female adult within the growth period, which had same feed conditions.”
Have the authors used only females? If yes, why?
Yes, I used only females. Because intensive feeding with the same standard is adopted at present, the main cashmere-producing population is ewes.
6. It is not clear if the samples were used to perform qRT-PCR. In my understanding thesse samples were collected to be used in qRT-PCR.
Yes, the samples were collected to be used in qRT-PCR.
7. Elucidate the number of females and males and clarify if they were also adults. Have the authors used only females? If yes, why?
the three female adult LCGs with coarse-type skin in anagen stage and three female adult LCGs with fine-type skins in anagen stage were collected for qRT-PCR.
Yes, they are all females, in the production they are all basic ewes.
8. Add "library preparation" between "RNA extraction and sequencing"
I have added it.
9. Please, determine the city, state and country of the manufacturer
I have add Takara kit (Dalian, China).
10. Replace "volume"with "amount"
I have replaced it.
11. Replace "RNA preparation" with "RNA library preparation"
I have replaced it.
12. Describe how the libaries were performed and which kit was used for.
Firstly, ribosomal RNA was removed using the Epicentre Ribo-zero rRNA Removal Kit (Epicentre, USA), and the rRNA was cleaned up by ethanol precipita_tion. Subsequently, in total two libraries were generated from the rRNA-depleted RNA using the NEBNext Ultra Dir_ectional RNA Library Prep Kit for Illumina (NEB, USA) following the manufacturer’s recommendations.
13. Add information about the number of yielded cDNA libraries that was used for sequencing.
Two libraries preparations were sequenced on an Illumina Hiseq 2000 platform.
14. Quote the authors of the algorithm FastQC
I have add reference [16].
15. While the authors said that filtered low quality reads and adaptors, they have not mention how. This information is needed.
I have add : All later analysis was based on the high-quality data.
16. Replace "the original sequencing data" with "the quality of the raw read data".
I have replaced it.
17. replace "adopted" with "used"
I have replaced it.
18. delete and only clean reads was retained
I have deleted only clean reads was retained.
19. You must quote preferentially the authors or alternatively the website address to all software or algorithms that you have used in this work
I have add reference [17].
20. Please, mentione the information about the reference genome: the species and their respective authors and bibliography or the public database link, where the genome data are available.
The reference genome is “https://www.ncbi.nlm.nih.gov/genome/?term=Capra+hircus”.
21. You must quote preferentially the authors or alternatively the website address to all software or algorithms that you have used in this work
I have add reference [18]
22. You must quote preferentially the authors or alternatively the website address to all software or algorithms that you have used in this work
The packages Coding Potential Calculator (CPC), Pfam-scan (PFAM), phylogenetic codonsubstitution frequency (phyloCSF), Coding-Non-Coding-Index (CNCI) are used to identify potential coding transcripts, and notnecessarilly filter out (only) lncRNAs as mentioned on line 86. The text should be rewritten
I have add reference [19], [20], [21], [22] and modified the sentence.
The class-code information originating from Cuffcompare was used to screen the candidate transcripts. Finally, transcripts that pass the protein-coding-score test were retained using CNCI [19], CPC [20], PFAM [21], and phyloCSF [22].
23. You must quote preferentially the authors or alternatively the website address to all software or algorithms that you have used in this work
I have add Pfam database (http://pfam.xfam.org/).
24. You must quote preferentially the authors or alternatively the website address to all software or algorithms that you have used in this work
The Figure 2A shows that the authors used FPKM instead of RPKM like written in the text.This is really odd, as edgeR does not converts to FPKM or RPKM. Actually the package requires the raw reads count and not normalized values such as RPKM, so the authors need leave it clear which data they used as input to the DE analysis. If they used RPKMor FPKM, the assumptions of the algorithm were violated and theanalyses should be re-done
I have add reference [23] and modified the sentence.
Cufflinks was used to calculate the fragments per kb per million reads (FPKM) for both lncRNAs and coding genes in each sample. Differentially expressed mRNAs and lncRNAs between the two libraries were identified by edgeR with raw read counts [23].
I'm sorry, due to my personal mistake, I have changed all the RPKM to FPKM in this article.
25. specify the manufacturer and additional information related to
I have add reference [14] and modified the sentence.
The GAPDH gene was used to standardize the expression level of quantitative real-time PCR [14],
26. Primer Express 5.0.
I have changed it to Primer 5.0.
27. quote the bibliography reference
I have add reference [25]
28. You must quote preferentially the authors or alternatively the website address to all software or algorithms that you have used in this work
I have add reference [26]
29. replace "between" with "in"
I have replaced it.
30. Replace "skin samples" with "obtained for skin samples of LCG and MCG species, respectively.
I have replaced it.
31. add (LCG) after 128,989,956
I have added it.
32. delete ", respectively"
I have deleted it.
33. Add the name of the species with the genome described, which was used in this work.
E.g. on the genome of ??????
I have added the name of the species.
34. delete via RNA-seq
I have deleted it.
34. Reword as suggested:
... to those reported by privious studies (REFERENCES).
I have added it.
35. change the " , " by a dot and start a new sentence as suggested below:
"...conservation scores (Figure 2B). The sequence results of 140 lncRNAs showed a certain degree of site conservation among the species, with regard to the mRNAs.
I have changed it as mentioned above.
The sequence results of lncRNAs showed a certain degree of site conservation among the species, with regard to the mRNAs.
35. Figure 2 needs to be improved in resolution, and quality of information. Add the unit of the lenght in the graphics
The pictures have been replaced and the units are written in legends.
38. Reword the sentence as suggested below:
We found 93 lncRNAs and 3084 mRNAs differentially expressed between LCG and MCG skin tissues.
I have modified it.
We found 93 lncRNAs and 3084 mRNAs differentially expressed between LCG and MCG skin tissues (q-value ≤0.05 and |log2FoldChange| ≥1).
41. Part of this sentence is related to methodology. You must focusing on the result obtained in the Result section. In this why, I suggest reword the sentence describing the results from the Figure 3A, and if possible explaining why the total number of DE lnc and m RNAs (3177) shown in Figure 3A is different from the total overlapping target genes. This is not well explained in the text. Results are not weel descibed here. Moreover, figure 3B is not necessary if the obtained results were clearly described in the text.
We performed an analysis on the target genes of lncRNAs and differentially expressed mRNAs, a total of 149 common genes were obtained, including KRT35, KRT32, NFKBIA.
Change 2199 to 2119 in 157 lines.
42. Figure 3 is not necessary to be presented in the article. I suggest describing properly the results obtained to such analyses in the tet.
The figure 3B has been deleted.
43. Replace "from using GO analysis" with "of GO"
I have replaced it.
44. delete " , respectively"
I have deleted it.
45. delete "via GO"
I have deleted it.
46. add "in the GO analysis" before the comma
I haved added it.
47. All legend texts need to be improved. The lengeds are very concise and not self-explanatory.
I have done it.
48. Is this result relevant to the context of the work? I'm not sure if it's a reliable result or an artifact.
Is a KEGG pathway,our related results also enriched this pathway
49. Figure 7 was not mentioned in the text
Previously, it is in the discussion,now it has been changed into Figure 4A.
50. Figure 6 quality is really poor, the gene names are not readable. There is no explanation on the light blue polygons.
Figure have been replaced, and legend has been changed
51.Reword this sentence as suggested: RNA-seq allows large-scale data production via high throughput sequencing, which has...
I have reworded it.
52. replace "find" with "investigate"
I have replaced it.
53. Reword the sentence as suggested:
Moreover, RNA-seq data can be used for feasible large-scale expression studies as well as for descovering class of biologically significant RNA transcripts, and characterization of lncRNAs ...
I haved reworded it.
54. The nine lncRNAs that were randomly selected to be used ...
I haved reworded it.
55. Reword as suggested:
allowed us to came to the following findings
I have reworded it.
55. Reword as suggested:
A prior study has reported that...
I have reworded it.
57. replace "showed with "shows"
I have replaced it.
58. Reword as suggested:
Here, we predicted...
I have reworded it.
59. Such information is already described in the results and here it must be discussed.
I have quoted a Reference.
Target genes KRT32, KRT35, KRT36, KRT38 of XLOC_008679 and target genes PSMA6, NFKBIA, KIAA0391 of XLOC_011319 were considered as a key candidate gene for cashmere fineness[34,41].
60. Replace with DSG4
I have replaced it.
61. replace "expressed" with "found"
I have replaced it.
62. Replace "data" with "study"
I have replaced it.
63. add also after "be"
I have added it.
64. These sentences are merely descriptive. There is no a relevant discussion linking they to the findings . Please, reword this part of the discussion, focusing on relevant insights .
I have reworded it.
The PPAR pathway may play a role during the start of secondary hair follicle development [24], target gene SORBS1 and UBC were enriched to this pathway in our data. Wnt signaling is needed for the establishment of hair follicles, and it is up-regulated at telogen and facilitated into antigens, which mainly results in the activation of bulge stem cells for the promotion of warding hair formation [65]. We found differentially expressed genes WNT11, BAMBI, WIF1, PPARD, and MMP7 were also enriched to Wnt signaling pathway. Target gene NFKBIA of XLOC_011060 and XLOC_011319 was enriched to NF-kappa B signaling pathway, which was reported may regulat hair follicle induction. The EDA/EDAR/NF-kB pathway is important for the development of hair follicles and epidermal appendages [66-69]. The lncRNA MTC promotes the regulation of hair follicle development and cashmere growth by activating NF-κB signaling [41].

Reviewer 2 Report
The manuscript “An Integrated analysis of cashmere-fineness lncRNAs in cashmere goats¨ by Zheng and collaborators seeks to identify lncRNAs involved in cashmere fineness in skin samples of Liaoning cashmere goats (LCG) and Inner Mongolia cashmere goats (MCG). To do that, RNA-seq. The article is well written and the problem is clearly discussed in the introduction. However, some of the methodology is incomplete or needs to be rewritten/ re-analysed:
-While the authors said that filtered low quality reads and adaptors, they have not mention how. This information is needed.
-The packages Coding Potential Calculator (CPC), Pfam-scan (PFAM), phylogenetic codonsubstitution frequency (phyloCSF), Coding-Non-Coding-Index (CNCI) are used to identify potential coding transcripts, and notnecessarilly filter out (only) lncRNAs as mentioned on line 86. The text should be rewritten.
- The Figure 2A shows that the authors used FPKM instead of RPKM like written in the text.This is really odd, as edgeR does not converts to FPKM or RPKM. Actually the package requires the raw reads count and not normalized values such as RPKM, so the authors need leave it clear which data they used as input to the DE analysis. If they used RPKMor FPKM, the assumptions of the algorithm were violated and theanalyses should be re-done.
Actually the whole figure 2 seems unnecessary, my suggestion is move it as supplementary material.
- Figure 6 quality is really poor, the gene names are not readable. There is no explanation on the light blue polygons.
- It's not clear what the authors want to show on Fig7.
- The GO analysis was not performed properly. There is no biological sense in obtaining common enriched paths for such different biological systems altogether, as DE genes are different for each of them. The correct analysis should be performed using the upregulated genes for each separate lineage, so the authors should have different enriched paths for each of them to do a proper discussion.
The discussion needs to be improved, is very descriptive.
Author Response
1. While the authors said that filtered low quality reads and adaptors, they have not mention how. This information is needed.
I have modified the sentence: FastQC was used to evaluate the quality of the raw read data [16]. Reads exceeding 10% unknown bases, reads containing adapters, and reads with low-quality bases were removed by FastQC. Simultaneously, a check on clean reads (Q20, Q30 and GC content) was performed. All later analysis was based on the high-quality data.
2. The packages Coding Potential Calculator (CPC), Pfam-scan (PFAM), phylogenetic codonsubstitution frequency (phyloCSF), Coding-Non-Coding-Index (CNCI) are used to identify potential coding transcripts, and notnecessarilly filter out (only) lncRNAs as mentioned on line 86. The text should be rewritten.
I have add reference [19], [20], [21], [22] and modified the sentence.
The class-code information originating from Cuffcompare was used to screen the candidate transcripts. Finally, transcripts that pass the protein-coding-score test were retained using CNCI [19], CPC [20], PFAM [21], and phyloCSF [22].
3. The Figure 2A shows that the authors used FPKM instead of RPKM like written in the text.This is really odd, as edgeR does not converts to FPKM or RPKM. Actually the package requires the raw reads count and not normalized values such as RPKM, so the authors need leave it clear which data they used as input to the DE analysis. If they used RPKMor FPKM, the assumptions of the algorithm were violated and theanalyses should be re-done.
I have add reference [23] and modified the sentence.
Cufflinks was used to calculate the fragments per kb per million reads (FPKM) for both lncRNAs and coding genes in each sample. Differentially expressed mRNAs and lncRNAs between the two libraries were identified by edgeR with raw read counts [23].
I'm sorry, due to my personal mistake, I have changed all the RPKM to FPKM in this article.
4. Actually the whole figure 2 seems unnecessary, my suggestion is move it as supplementary material.
I've replaced the clear pictures. If you think it's necessary, I can put it in the supplement.
5. Figure 6 quality is really poor, the gene names are not readable. There is no explanation on the light blue polygons.
Figure 6 have been replaced, I'm very sorry I just found that the legend hasn't been changed.
6. It's not clear what the authors want to show on Fig7.
The GO analysis was not performed properly. There is no biological sense in obtaining common enriched paths for such different biological systems altogether, as DE genes are different for each of them. The correct analysis should be performed using the upregulated genes for each separate lineage, so the authors should have different enriched paths for each of them to do a proper discussion.
Our Figure 7 may not be very clear. We can change Figure 7 to Table 4, now the Figure 7 at the position is Figure 4A. Can we all keep it (Figure 4A and Table 4)? (check author_response.docx)
Table 4. GO term of cashmere fineness
GO term | lncRNA | Up-Target gene | Down- Target gene |
intermediate filament | XLOC_008679 | ___ | LOC102179515, KRT32, KRT36, KRT35, LOC102176457, LOC102176457, KRT38, LOC100861381, LOC102179881 |
intermediate filament cytoskeleton | XLOC_008679 | ___ | LOC102179515, KRT32, KRT36, KRT35, LOC102176457, LOC102176457, KRT38, LOC100861381, LOC102179881 |
viral capsid | XLOC_008679
| ___ | KRT35, KRT36, LOC100861381, LOC102179515, KRT32, KRT38 |
XLOC_019874 | RMND5B | ||
XLOC_020860 | GOPC | ||
XLOC_010504 | LOC102170006, PLA2G2A | ||
XLOC_001617 | LEKR1 | ||
XLOC_015428 | IQGAP3 | ||
XLOC_006506 | LOC102177360 |
7. The discussion needs to be improved, is very descriptive.
The discussion section has been revised.
I changed “We searched the coding genes of lncRNAs, respectively ranging from 10 kb/100 kb upstream and downstream, taking them as cis target genes. We made a prediction for the functions of lncRNAs on goat skin, and determined as a result that mRNAs can act with lncRNAs.” to “Target genes KRT32, KRT35, KRT36, KRT38 of XLOC_008679 and target genes PSMA6, NFKBIA, KIAA0391 of XLOC_011319 were considered as a key candidate gene for cashmere fineness[34,41].”
I changed “The PPAR pathway may play a role during the start of secondary hair follicle development [24], and this was also enriched in our data. Wnt signaling is needed for the establishment of hair follicles, and it is up-regulated at telogen and facilitated into antigens, which mainly results in the activation of bulge stem cells for the promotion of warding hair formation [53]; these relayed signals have a close relation to ?-catenin and Lef1 [54,55]. The EDA/EDAR/NF-kB pathway is important for the development of hair follicles and epidermal appendages [56-59]. The lncRNA MTC promotes the regulation of hair follicle development and cashmere growth by activating NF-κB signaling [60]. Adenosine-mediated signaling pathways were involved in minoxidil-induced hair growth [61]. Sphingolipid metabolites are involved in cell survival and death; Sph itself induces hair cell death and increase hair cell loss induced via CDDP [62]. The effect of lncRNAs on hair development has been further clarified.” to “The PPAR pathway may play a role during the start of secondary hair follicle development [24], target gene SORBS1 and UBC were enriched to this pathway in our data. Wnt signaling is needed for the establishment of hair follicles, and it is up-regulated at telogen and facilitated into antigens, which mainly results in the activation of bulge stem cells for the promotion of warding hair formation [65]. We found differentially expressed genes WNT11, BAMBI, WIF1, PPARD, and MMP7 were also enriched to Wnt signaling pathway. Target gene NFKBIA of XLOC_011060 and XLOC_011319 was enriched to NF-kappa B signaling pathway, which was reported may regulat hair follicle induction. The EDA/EDAR/NF-kB pathway is important for the development of hair follicles and epidermal appendages [66-69]. The lncRNA MTC promotes the regulation of hair follicle development and cashmere growth by activating NF-κB signaling [41]. We think that NF-kappa B signaling pathway may be related to cashmere fineness. ”
